# Semisupervised Clustering, AND-Queries and Locally Encodable Source Coding

**Arya Mazumdar**
College of Information & Computer Sciences
University of Massachusetts Amherst
Amherst, MA 01003
arya@cs.umass.edu

**Soumyabrata Pal**
College of Information & Computer Sciences
University of Massachusetts Amherst
Amherst, MA 01003
soumyabratap@umass.edu

## Abstract

Source coding is the canonical problem of data compression in information theory. In a *locally encodable* source coding, each compressed bit depends on only few bits of the input. In this paper, we show that a recently popular model of semisupervised clustering is equivalent to locally encodable source coding. In this model, the task is to perform multiclass labeling of unlabeled elements. At the beginning, we can ask in parallel a set of simple queries to an oracle who provides (possibly erroneous) binary answers to the queries. The queries cannot involve more than two (or a fixed constant number $\Delta$ of) elements. Now the labeling of all the elements (or clustering) must be performed based on the (noisy) query answers. The goal is to recover all the correct labelings while minimizing the number of such queries. The equivalence to locally encodable source codes leads us to find lower bounds on the number of queries required in variety of scenarios. We are also able to show fundamental limitations of pairwise 'same cluster' queries - and propose pairwise AND queries, that provably performs better in many situations.

## 1 Introduction

Suppose we have $n$ elements, and the $i$th element has a label $X_i \in \{0, 1, \ldots, k-1\}, \forall i \in \{1, \ldots, n\}$. We consider the task of learning the labels of the elements (or learning the label vector). This can also be easily thought of as a clustering problem of $n$ elements into $k$ clusters, where there is a ground-truth clustering[1]. There exist various approaches to this problem in general. In many cases some similarity values between pair of elements are known (a high similarity value indicate that they are in the same cluster). Given these similarity values (or a weighted complete graph), the task is equivalent to to graph clustering; when perfect similarity values are known this is equivalent to finding the connected components of a graph.

A recent approach to clustering has been via crowdsourcing. Suppose there is an oracle (expert labelers, crowd workers) with whom we can make pairwise queries of the form "do elements $u$ and $v$ belong to the same cluster?". We will call this the 'same cluster' query (as per [4]). Based on the answers from the oracle, we then try to reconstruct the labeling or clustering. This idea has seen a recent surge of interest especially in the entity resolution research (see, for e.g. [33, 30, 8, 20]). Since each query to crowd workers cost time and money, a natural objective will be to minimize the number of queries to the oracle and still recover the clusters exactly. Carefully designed adaptive and interactive querying algorithms for clustering has also recently been developed [33, 30, 8, 22, 21]. In

particular, the query complexity for clustering with a $k$-means objective had recently been studied in [4], and there are significant works in designing optimal crowdsourcing schemes in general (see, [12, 13, 28, 34, 15]). Note that, a crowd worker may potentially handle more than two elements at a time; however it is of interest to keep the number of elements involved in a query as small as possible. As an example, recent work in [31] considers triangle queries (involving three elements in a query). Also crowd workers can compute some simple functions on this small set of inputs - instead of answering a 'same cluster' query. But again it is desirable that the answer the workers provide to be simple, such as a binary answer.

The queries to the oracle can be asked adaptively or non-adaptively. For the clustering problem, both the adaptive version and the nonadaptive versions have been studied. While both versions have obvious advantages and disadvantages, for crowdsourcing applications it is helpful to have a parallelizable querying scheme in most scenarios for faster response-rate and real time analysis. In this paper, we concentrate on the nonadaptive version of the problem, i.e., we perform the labeling algorithm after all the query answers are all obtained.

Budgeted crowdsourcing problems can be quite straight-forwardly viewed as a canonical source-coding or source-channel coding problem of information theory (e.g., see the recent paper [14]). A main contribution of our paper is to view this as a *locally encodable* source coding problem: a data compression problem where each compressed bit depends only on a constant number of input bits. The notion of locally encodable source coding is not well-studied even within information theory community, and the only place where it is mentioned to the best of our knowledge is in [23], although the focus of that paper is a related notion of *smooth* encoding. Another related notion of *local decoding* seem to be much more well-studied [19, 18, 16, 26, 6, 25, 5, 32].

By posing the querying problem as such we can get a number of information theoretic lower bounds on the number of queries required to recover the correct labeling. We also provide nonadaptive schemes that are near optimal. Another of our main contributions is to show that even within queries with binary answers, 'same cluster' queries (or XOR queries) may not be the best possible choice. A smaller number of queries can be achieved for approximate recovery by using what we call an AND query. Among our settings, we also consider the case when the oracle gives incorrect answers with some probability. A simple scheme to reduce errors in this case could be to take a majority vote after asking the same question to multiple different crowd workers. However, often that is not sufficient. Experiments on several real datasets (see [21]) with answers collected from Amazon Mechanical Turk [9, 29] show that majority voting could even increase the errors. Interestingly, such an observation has been made by a recent paper as well [27, Figure 1]. The probability of error of a query answer may also be thought of as the aggregated answer after repeating the query several times. Once the answer has been aggregated, it cannot change – and thus it reduces to the model where repeating the same question multiple times is not allowed. On the other hand, it is usually assumed that the answers to different queries are independently erroneous (see [10]). Therefore we consider the case where repetition of a same query multiple times is not allowed[2], however different queries can result in erroneous answers independently.

In this case, the best known algorithms need $O(n \log n)$ queries to perform the clustering with two clusters [21]. We show that by employing our AND querying method $(1 - \delta)$-proportion of all labels in the label vector will be recovered with only $O(n \log \frac{1}{\delta})$ queries.

Along the way, we also provide new information theoretic results on fundamental limits of locally encodable source coding. While the the related notion of locally decodable source code [19, 16, 26, 6], as well as smooth compression [23, 26] have been studied, there was no nontrivial result known related to locally encodable codes in general. Although the focus of this paper is primarily theoretical, we also perform a real crowdsourcing experiment to validate our algorithm.

## 2  Problem Setup and Information Theoretic View

For $n$ elements, consider a label vector $\boldsymbol{X} \in \{0, \ldots, k-1\}^n$, where $X_i$, the $i$th entry of $\boldsymbol{X}$, is the label of the $i$th element and can take one of $k$ possible values. Suppose $P(X_i = j) = p_j \forall j$ and $X_i$'s are independent. In other words, the prior distribution of the labels is given by the vector

$\boldsymbol{p} \equiv (p_0, \ldots, p_{k-1})$. For the special case of $k = 2$, we denote $p_0 \equiv 1 - p$ and $p_1 \equiv p$. While we emphasize on the case of $k = 2$ our results extends in the case of larger $k$, as will be mentioned.

A query $Q : \{0, \ldots, k - 1\}^\Delta \to \{0, 1\}$ is a deterministic function that takes as argument $\Delta$ labels, $\Delta \ll n$, and outputs a binary answer. While the query answer need not be binary, we restrict ourselves mostly to this case for being the most practical choice.

Suppose a total of $m$ queries are made and the query answers are given by $\boldsymbol{Y} \in \{0, 1\}^m$. The objective is to reconstruct the label vector $\boldsymbol{X}$ from $\boldsymbol{Y}$, such that the number of queries $m$ is minimized.

We assume our recovery algorithms to have the knowledge of $\boldsymbol{p}$. This prior distribution, or the relative sizes of clusters, is usually easy to estimate by subsampling a small ($O(\log n)$) subset of elements and performing a complete clustering within that set (by, say, all pairwise queries). In many prior works, especially in the recovery algorithms of popular statistical models such as stochastic block model, it is assumed that the relative sizes of the clusters are known (see [1]).

We also consider the setting where query answers may be erroneous with some probability of error. For crowdsourcing applications, this is a valid assumption since many times even expert labelers can make errors, and such assumption can be made. To model this we assume each entry of $\boldsymbol{Y}$ is flipped independently with some probability $q$. Such independence assumption has been used many times previously to model errors in crowdsourcing systems (see, e.g., [10]). While this may not be the perfect model, we *do not allow a single query to be repeated multiple times in our algorithms* (see the Introduction for a justification). For the analysis of our algorithm we just need to assume that the answers to different queries are independent. While we analyze our algorithms under these assumptions for theoretical guarantees, it turns out that even in real crowdsourcing situations our algorithmic results mostly follow the theoretical results, giving further validation to the model.

For the $k = 2$ case, and when $q = 0$ (perfect oracle), it is easy to see that $n$ queries are sufficient for the task. One simply compares every element with the first element. This does not extend to the case when $k > 2$: for perfect recovery, and without any prior, one must make $O(n^2)$ queries in this case. When $q > 0$ (erroneous oracle), it has been shown that a total number of $O(\gamma n k \log n)$ queries are sufficient [21], where $\gamma$ is the ratio of the sizes of the largest and smallest clusters.

**Information theoretic view.** The problem of learning a label vector $\boldsymbol{x}$ from queries is very similar to the canonical source coding (data compression) problem from information theory. In the source coding problem, a (possibly random) vector $\boldsymbol{X}$ is 'encoded' into a usually smaller length binary vector called the *compressed vector*[3] $\boldsymbol{Y} \in \{0, 1\}^m$. The decoding task is to again obtain $\boldsymbol{X}$ from the compressed vector $\boldsymbol{Y}$. It is known that if $\boldsymbol{X}$ is distributed according to $\boldsymbol{p}$, then $m \approx nH(\boldsymbol{p})$ is both necessary and sufficient to recover $\boldsymbol{x}$ with high probability, where $H(\boldsymbol{p}) = -\sum_i p_i \log p_i$ is the entropy of $\boldsymbol{p}$.

We can cast our problem in this setting naturally, where entries of $\boldsymbol{Y}$ are just answers to queries made on $\boldsymbol{X}$. The main difference is that in source coding each $Y_i$ may potentially depend on all the entries of $\boldsymbol{X}$; while in the case of label learning, each $Y_i$ may depend on only $\Delta$ of the $x_i$s.

We call this *locally encodable source coding*. This terminology is analogous to the recently developed literature on locally decodable source coding [19, 16]. It is called locally encodable, because each compressed bit depend only on $\Delta$ of the source (input) bits. For locally decodable source coding, each bit of the reconstructed sequence $\hat{\boldsymbol{X}}$ depends on at most a prescribed constant number $\Delta$ of bits from the compressed sequence. Another closely related notion is that of 'smooth compression', where each source bit contributes to at most $\Delta$ compressed bits [23]. Indeed, in [23], the notion of locally encodable source coding is also present where it was called robust compression. We provide new information theoretic lower bounds on the number of queries required to reconstruct $\boldsymbol{X}$ exactly and approximately for our problem.

For the case when there are only two labels, the 'same cluster' query is equivalent to an Boolean XOR operation between the labels. There are some inherent limitations to these functions that prohibit the 'same cluster' queries to achieve the best possible number of queries for the 'approximate' recovery of labeling problem. We use an old result by Massey [17] to establish this limitation. We show that, instead using an operation like Boolean AND, much smaller number of queries are able to recover most of the labels correctly.

We also consider the case when the oracle gives faulty answer, or $\boldsymbol{Y}$ is corrupted by some noise (the *binary symmetric channel*). This setting is analogous to the problem of *joint source-channel coding*. However, just like before, each encoded bit must depend on at most $\Delta$ bits. We show that for the approximate recovery problem, AND queries are again performing substantially well. In a real crowdsourcing experiment, we have seen that if crowd-workers have been provided with the same set of pairs and being asked for 'same cluster' queries as well as AND queries, the error-rate of AND queries is lower. The reason is that for a correct 'no' answer in an AND query, a worker just need to know one of the labels in the pair. For a 'same cluster' query, both the labels must be known to the worker for any correct answer.

There are multiple reasons why one would ask for a 'combination' or function of multiple labels from a worker instead of just asking for a label itself (a 'label-query'). Note that, asking for labels will never let us recover the clusters in less than $n$ queries, whereas, as we will see, the queries that combine labels will. Also in case of erroneous answer with AND queries or 'same cluster' queries, we have the option of not repeating a query, and still reduce errors. No such option is available with direct label-queries.

**Contributions.** In summary our contributions can be listed as follows.

1. Noiseless queries and exact recovery (Sec. 3.1): For two clusters, we provide a querying scheme that asks $\alpha n, \alpha < 1$ number of nonadaptive pairwise 'same cluster' queries, and recovers the all labels with high probability, for a range of prior probabilities. We also provide a new lower bound that is strictly better than $nH(\boldsymbol{p})$ for some $\boldsymbol{p}$.

2. Noiseless queries and approximate recovery (Sec. 3.2): We provide a new lower bound on the number of queries required to recover $(1 - \delta)$ fraction of the labels $\delta > 0$. We also show that 'same cluster' queries are insufficient, and propose a new querying strategy based on AND operation that performs substantially better.

3. Noisy queries and approximate recovery (Sec. 3.3). For this part we assumed the query answer to be $k$-ary ($k \geq 2$) where $k$ is the number of clusters. This section contains the main algorithmic result that uses the AND queries as main primitive. We show that, even in the presence of noise in the query answers, it is possible to recover $(1 - \delta)$ proportion of all labels correctly with only $O(n \log \frac{k}{\delta})$ nonadaptive queries. We validate this theoretical result in a real crowdsourcing experiment in Sec. 4.

## 3   Main results and Techniques

### 3.1   Noiseless queries and exact recovery

In this scenario we assume the query answer from oracle to be perfect and we wish to get back the all of the original labels exactly without any error. Each query is allowed to take only $\Delta$ labels as input. When $\Delta = 2$, we are allowed to ask only pairwise queries. Let us consider the case when there are only two labels, i.e., $k = 2$. That means the labels $X_i \in \{0, 1\}, 1 \leq i \leq n$, are iid Bernoulli($p$) random variable. Therefore the number of queries $m$ that are necessary and sufficient to recover all the labels with high probability is approximately $nh(p) \pm o(n)$ where $h(x) \equiv -x \log_2 x - (1-x) \log_2(1-x)$ is the binary entropy function. However the sufficiency part here does not take into account that each query can involve only $\Delta$ labels.

**Querying scheme:** We use the following type of queries. For each query, labels of $\Delta$ elements are given to the oracle, and the oracle returns a simple XOR operation of the labels. Note, for $\Delta = 2$, our queries are just 'same cluster' queries.

**Theorem 1.** *There exists a querying scheme with $m = \frac{n(h(p)+o(1))}{\log_2 \frac{1}{\alpha}}$ queries of above type, where $\alpha = \frac{1}{2}(1 + (1 - 4p(1-p))^\Delta)$, such that it will be possible to recover all the labels with high probability by a Maximum Likelihood decoder.*

*Proof.* Let the number of queries asked is $m$. Let us define $\mathcal{Q}$ to be the random binary query matrix of dimension $m \times n$ where each row has exactly $\Delta$ ones, other entries being zero. Now for a label vector $\boldsymbol{X}$ we can represent the set of query outputs by $\boldsymbol{Y} = \mathcal{Q}\boldsymbol{X} \mod 2$. Now if we use Maximum Likelihood Decoding then we will not make an error as long as the query output vector is different

for every $X$ that belong to the *typical set*[4] of $\boldsymbol{X}$. Let us define a 'bad event' for two different label vectors $X^1$ and $X^2$ to be the event $\mathcal{Q}X^1 = \mathcal{Q}X^2$ or $\mathcal{Q}(X^1 + X^2) = 0 \mod 2$ because in that case we will not be able to differentiate between those two sequences. Now consider a random ensemble of matrices where in each row $\Delta$ positions are chosen uniformly randomly from the $n$ positions to be 1. In this random ensemble, the probability of a 'bad event' for any two fixed typical label vectors $X^1$ and $X^2$ is going to be

$$\left(\frac{\sum_{\substack{i=0:\Delta \\ i \text{ even}}} \binom{nr(p)}{i}\binom{n-nr(p)}{\Delta-i}}{\binom{n}{\Delta}}\right)^m \leq \left(\frac{\frac{1}{2}(\binom{n}{\Delta} + \binom{n-2nr(p)}{\Delta})}{\binom{n}{\Delta}}\right)^m \leq \left(\frac{1}{2}(1 + (1 - 2r(p))^\Delta)\right)^m,$$

where $r(p) = 2p(1-p)$. This is because , $X^1 + X^2 \mod 2$ has $r(p) = 2p(1-p)$ ones with high probability since they are typical vectors.

Now we have to use the 'coding theoretic' idea of expurgation to complete the analysis. From linearity of expectation, the expected number of 'bad events' is going to be

$$\binom{T}{2}\left(\frac{1}{2}(1 + (1 - 2r(p))^\Delta)\right)^m,$$

where $T$ is the size of the typical set and $T \leq 2^{n(h(p)+o(1))}$. If this expected number of 'bad events' is smaller than $\epsilon T$ then for every 'bad event', we can throw out 1 label vector and there will be no more bad events. This will imply perfect recovery, as long as

$$\binom{T}{2}\left(\frac{1}{2}(1 + (1 - 2r(p))^\Delta)\right)^m < \epsilon T.$$

Substituting the upper bound for $T$, we have that perfect recovery is possible as long as, $\frac{m}{n} > (h(p) + o(1) - \frac{\log 2\epsilon}{n})/(\log \frac{1}{\alpha})$. Now if we take $\epsilon$ to be of the form $n^{-\beta}$ for $\beta > 0$ then asymptotically we will have a vanishing fraction of typical label vectors which will be expurgated and $\frac{\log \epsilon}{n} \to 0$. Therefore $m = \frac{n(h(p)+o(1))}{\log \frac{1}{\alpha}}$ queries will going to recover all the labels with high probability. Hence there must exist a querying scheme with $m = \frac{n(h(p)+o(1))}{\log \frac{1}{\alpha}}$ queries that will work. $\qquad \square$

The number of sufficient queries guaranteed by the above theorem is strictly less than $n$ for all $0.0694 \leq p < 0.5$ even for $\Delta = 2$. Note that, with $\Delta = 2$, by querying the first element with all others nonadaptively (total $n - 1$ queries), it is possible to deduce the two clusters. In contrast, if one makes just random 'same cluster' queries, then $O(n \log n)$ queries are required to recover the clusters with high probability (see, e.g., [2]).

Now we provide a lower bound on the number of queries required.

**Theorem 2.** *The minimum number of queries necessary to recover all labels with high probability is at least by $nh(p) \cdot \max\{1, \max_\rho \frac{(1-\rho)}{h(\frac{(1-\rho)r(p)\Delta}{\rho})}\}$ where $r(p) \equiv 2p(1-p)$.*

*Proof.* Every query involves at most $\Delta$ elements. Therefore the average number of queries an element is part of is $\frac{\Delta m}{n}$. Therefore $1 - \rho$ fraction of all the elements (say the set $S \subset \{1, \ldots, n\}$) are part of less than $\frac{\Delta m}{\rho n}$ queries. Now consider the set $\{1, \ldots, n\} \setminus S$. Consider all typical label vectors $\mathcal{C} \in \{0, 1\}^n$ such that their projection on $\{1, \ldots, n\} \setminus S$ is a fixed typical sequence. We know that there are $2^{n(1-\rho)h(p)}$ such sequences. Let $\boldsymbol{X}_0$ be one of these sequences. Now, almost all sequences of $\mathcal{C}$ must have a distance of $n(1-\rho)r(p)+o(n)$ from $\boldsymbol{X}_0$. Let $\boldsymbol{Y}_0$ be the corresponding query outputs when $\boldsymbol{X}_0$ is the input. Now any query output for input belonging to $\mathcal{C}$ must reside in a Hamming ball of radius $\frac{(1-\rho)r(p)\Delta m}{\rho}$ from $\boldsymbol{Y}_0$. Therefore we must have $mh(\frac{(1-\rho)r(p)\Delta}{\rho}) \geq n(1-\rho)h(p)$. $\quad \square$

This lower bound is better than the naive $nh(p)$ for $p < 0.03475$ when $\Delta = 2$.

For $\Delta = 2$, the plot of the corresponding upper and lower bounds have been shown in Figure 1. The main takeaway from this part is that, by exploiting the prior probabilities (or relative cluster sizes), it is possible to know the labels with strictly less than $n$ queries (and close to the lower bound for $p \geq 0.3$), even with pairwise 'same cluster' queries.

## 3.2 Noiseless queries and approximate recovery

Again let us consider the case when $k = 2$, i.e., only two possible labels. Let $\boldsymbol{X} \in \{0, 1\}^n$ be the label vector. Suppose we have a querying algorithm that, by using $m$ queries, recovers a label vector $\hat{\boldsymbol{X}}$.

**Definition.** We call a querying algorithm to be $(1 - \delta)$-good if for any label vector, at least $(1 - \delta)n$ labels are correctly recovered. This means for any label-recovered label pair $X, \hat{X}$, the Hamming distance is at most $\delta n$. For an almost equivalent definition, we can define a distortion function $d(X, \hat{X}) = X + \hat{X} \mod 2$, for any two labels $X, \hat{X} \in \{0, 1\}$. We can see that $\mathbb{E}d(X, \hat{X}) = \Pr(X \neq \hat{X})$, which we want to be bounded by $\delta$.

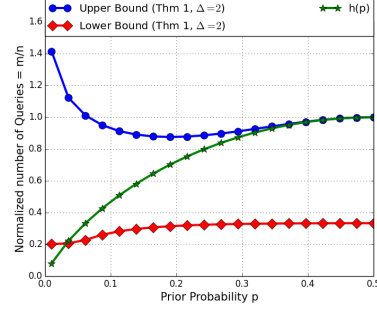

Figure 1: Number of pairwise queries for noiseless queries and exact recovery

Using standard rate-distortion theory [7], it can be seen that, if the queries could involve an arbitrary number of elements then with $m$ queries it is possible to have a $(1 - \tilde{\delta}(m/n))$-good scheme where $\tilde{\delta}(\gamma) \equiv h^{-1}(h(p) - \gamma)$. When each query is allowed to take only at most $\Delta$ inputs, we have the following lower bound for $(1 - \delta)$-good querying algorithms.

**Theorem 3.** *In any $(1 - \delta)$-good querying scheme with $m$ queries where each query can take as input $\Delta$ elements, the following must be satisfied (below $h'(x) = \frac{dh(x)}{dx}$):*

$$\delta \geq \tilde{\delta}\left(\frac{m}{n}\right) + \frac{h(p) - h(\tilde{\delta}(\frac{m}{n}))}{h'(\tilde{\delta}(\frac{m}{n}))(1 + e^{\Delta h'(\tilde{\delta}(\frac{m}{n}))})}$$

The proof of this theorem is quite involved, and we have included it in the appendix in the supplementary material.

One of the main observation that we make is that the 'same cluster' queries are highly inefficient for approximate recovery. This follows from a classical result of Ancheta and Massey [17] on the limitation of linear codes as rate-distortion codes. Recall that, the 'same cluster' queries are equivalent to XOR operation in the binary field, which is a linear operation on $GF(2)$. We rephrase a conjecture by Massey in our terminology.

**Conjecture 1** ('same cluster' query lower bound). *For any $(1 - \delta)$-good scheme with $m$ 'same cluster' queries $(\Delta = 2)$, the following must be satisfied: $\delta \geq p(1 - \frac{m}{nh(p)})$.*

This conjecture is known to be true at the point $p = 0.5$ (equal sized clusters). We have plotted these two lower bounds in Figure 2.

Now let us provide a querying scheme with $\Delta = 2$ that will provably be better than 'same cluster' schemes.

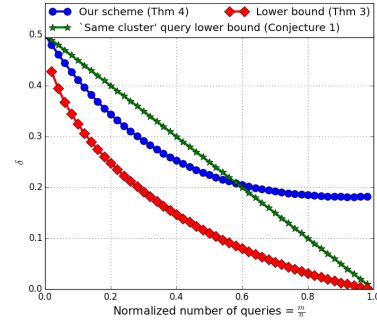

Figure 2: Performance of $(1 - \delta)$-good schemes with noiseless queries; $p = 0.5$

**Querying scheme: AND Queries:** We define the AND query $Q : \{0, 1\}^2 \to \{0, 1\}$ as $Q(X, X') = X \bigwedge X'$, where $X, X' \in \{0, 1\}$, so that $Q(X, X') = 1$ only when both the elements have labels equal to 1. For each pairwise query the oracle will return this AND operation of the labels.

**Theorem 4.** *There exists a $(1 - \delta)$-good querying scheme with $m$ pairwise AND queries such that*

$$\delta = pe^{-\frac{2m}{n}} + \sum_{d=1}^{n} \frac{e^{-\frac{2m}{n}}(\frac{2m}{n})^d}{d!} \sum_{k=1}^{d} \binom{n}{k} \frac{f(k, d)}{n^d}(1 - p)^k p$$

*where $f(k,d) = \sum_{i=0}^{k}(-1)^i \binom{k}{i}(k-i)^d$.*

*Proof.* Assume $p < 0.5$ without loss of generality. Consider a random bipartite graph where each 'left' node represent an element labeled according to the label vector $\boldsymbol{X} \in \{0,1\}^n$ and each 'right' node represents a query. All the query answers are collected in $\boldsymbol{Y} \in \{0,1\}^m$. The graph has right-degree exactly equal to 2. For each query the two inputs are selected uniformly at random without replacement.

*Recovery algorithm:* For each element we look at the queries that involves it and estimate its label as 1 if any of the query answers is 1 and predict 0 otherwise. If there are no queries that involves the element, we simply output 0 as the label.

Since the average left-degree is $\frac{2m}{n}$ and since all the edges from the right nodes are randomly and independently thrown, we can model the degree of each left-vertex by a Poisson distribution with the mean $\lambda = \frac{2m}{n}$. We define element $j$ to be a two-hop-neighbor of $i$ if there is at least one query which involved both the elements $i$ and $j$ . Under our decoding scheme we only have an error when the label of $i$, $X_i = 1$ and the labels of all its two-hop-neighbors are 0. Hence the probability of error for estimating $X_i$ can be written as, $\Pr(X_i \neq \hat{X}_i) = \sum_d \Pr(\text{degree}(i) = d) \Pr(X_i \neq \hat{X}_i \mid \text{degree}(i) = d)$. Now let us estimate $\Pr(X_i \neq \hat{X}_i \mid \text{degree}(i) = d)$. We further condition the error on the event that there are $k$ distinct two-hop-neighbors (lets call the number of distinct neighbors of $i$ as $\text{Dist}(i)$) and hence we have that $\Pr(X_i \neq \hat{X}_i \mid \text{degree}(i) = d) = \sum_{k=1}^{d} \Pr(\text{Dist}(i) = k) \Pr(X_i \neq \hat{X}_i | \text{degree}(i) = d, \text{Dist}(i) = k) = \sum_{k=1}^{d} \binom{n}{k} \frac{f(k,d)}{n^d} p(1-p)^k$. Now using the Poisson assumption we get the statement of the theorem. $\qquad\square$

The performance of this querying scheme is plotted against the number of queries for prior probabilities $p = 0.5$ in Figure 2.

**Comparison with 'same cluster' queries:** We see in Figure 2 that the AND query scheme beats the 'same cluster' query lower bound for a range of query-performance trade-off in approximate recovery for $p = \frac{1}{2}$. For smaller $p$, this range of values of $\delta$ increases further. If we randomly choose 'same cluster' queries and then resort to maximum likelihood decoding (note that, for AND queries, we present a simple decoding) then $O(n \log n)$ queries are still required even if we allow for $\delta$ proportion of incorrect labels (follows from [11]). The best performance for 'same cluster' query in approximate recovery that we know of for small $\delta$ is given by: $m = n(1 - \delta)$ (neglect $n\delta$ elements and just query the $n(1 - \delta)$ remaining elements with the first element). However, such a scheme can be achieved by AND queries as well in a similar manner. Therefore, there is no point in the query vs $\delta$ plot that we know of where 'same cluster' query achievability outperforms AND query achievability.

### 3.3 Noisy queries and approximate recovery

This section contains our main algorithmic contribution. In contrast to the previous sections here we consider the general case of $k \geq 2$ clusters. Recall that the label vector $\boldsymbol{X} \in \{0, 1, \dots, k-1\}^n$, and the prior probability of each label is given by the probabilities $\boldsymbol{p} = (p_0, \dots, p_{k-1})$. Instead of binary output queries, in this part we consider an oracle that can provide one of $k$ different answers. We consider a model of noise in the query answer where the oracle provides correct answer with probability $1 - q$, and any one of the remaining incorrect answers with probability $\frac{q}{k-1}$. Note that we do not allow the same query to be asked to the oracle multiple time (see Sec. 2 for justification). We also define a $(1 - \delta)$-good approximation scheme exactly as before.

**Querying Scheme:** We only perform pairwise queries. For a pair of labels $X$ and $X'$ we define a query $Y = Q(X, X') \in \{0, 1, \dots, k-1\}$. For our algorithm we define the $Q$ as

$$Q(X, X') = \left\{ \begin{array}{ll} i & if \quad X = X' = i \\ 0 & \quad\quad \text{otherwise.} \end{array} \right\}$$

We can observe that for $k = 2$, this query is exactly same as the binary AND query that we defined in the previous section. In our querying scheme, we make a total of $\frac{nd}{2}$ queries, for an integer $d > 1$. We design a $d$-regular graph $G(V, E)$ where $V = \{1, \dots, n\}$ is the set of elements that we need to label. We query all the pairs of elements $(u, v) \in E$.

Under this querying scheme, we propose to use Algorithm 1 for reconstructions of labels.

**Theorem 5.** *The querying scheme with $m = O(n \log \frac{k}{\delta})$ queries and Algorithm 1 is $(1 - \delta)$-good for approximate recovery of labels from noisy queries.*

---

**Algorithm 1** Noisy query approximate recovery with $\frac{nd}{2}$ queries

---

**Require:** PRIOR $\boldsymbol{p} \equiv (p_0, \ldots, p_{k-1})$
**Require:** Query Answers $Y_{u,v} : (u,v) \in E$
  **for** $i \in [1, 2, \ldots, k-1]$ **do**
    $C_i = \frac{dq}{k-1} + \frac{dp_i}{2}\left(1 - \frac{qk}{k-1}\right)$
  **end for**
  **for** $u \in V$ **do**
    **for** $i \in [1, 2, \ldots, k-1]$ **do**
      $N_{u,i} = \sum_{v=1}^{d} 1\{Y_{u,v} = i\}$
      **if** $N_{u,i} \geq \lceil C_i \rceil$ **then**
        $X_u \leftarrow i$
        Assigned $\leftarrow$ True
        **break**
      **end if**
    **end for**
    **if** $\neg$ Assigned **then**
      $X_u \leftarrow 0$
    **end if**
  **end for**

---

We can come up with more exact relation between number of queries $m = \frac{nd}{2}$, $\delta, p, q$ and $k$. This is deferred to the appendix in the supplementary material.

*Proof of Theorem 5.* The total number of queries is $m = \frac{nd}{2}$. Now for a particular element $u \in V$, we look at the values of $d$ noisy oracle answers $\{Y_{u,v}\}_{v=1}^{d}$. We have, $\mathbb{E}(N_{u,i}) = \frac{dq}{k-1} + dp_i\left(1 - \frac{qk}{k-1}\right)$ when the true label of $u$ is $i \neq 0$. When the true label is something else, $\mathbb{E}(N_{u,i}) = \frac{dq}{k-1}$. There is an obvious gap between these expectations. Clearly when the true label is $i$, the probability of error in assignment of the label of $u$ is given by, $P_i \leq \sum_{j:j\neq i, j\neq 0} \Pr(N_{u,j} > C_j) + \Pr(N_{u,i} < C_i) \leq cke^{-2d\epsilon^2}$ for some constants $c$ and $\epsilon$ depending on the gap, from Chernoff bound. And when the true label is $0$, the probability of error is $P_0 \leq \sum_{j:j\neq 0} P(N_{u,j} > C_j) \leq c'ke^{-2d\epsilon'^2}$, for some constants $c', \epsilon'$. Let $\delta = \sum_i p_i P_i$, we can easily observe that $d$ scales as $O(\log \frac{k}{\delta})$. Hence the total number of queries is $\frac{nd}{2} = O(n \log \frac{k}{\delta})$.

The only thing that remains to be proved is that the number of incorrect labels is $\delta n$ with high probability. Let $Z_u$ be the event that element $u$ has been incorrectly labeled. Then $\mathbb{E}Z_u = \delta$. The total number of incorrectly labeled elements is $Z = \sum_u Z_u$. We have $\mathbb{E}Z = n\delta$. Now define $Z_u \sim Z_v$ if $Z_u$ and $Z_v$ are dependent. Now $\Delta^* \equiv \sum_{Z_u \sim Z_v} \Pr(Z_u | Z_v) \leq d^2 + d$ because the maximum number of nodes dependent with $Z_u$ are its 1-hop and 2-hop neighbors. Now using Corollary 4.3.5 in [3], it is evident that $Z = \mathbb{E}Z = n\delta$ almost always. $\qquad\square$

The theoretical performance guarantee of Algorithm 1 (a detailed version of Theorem 5 is in the supplementary material) for $k = 2$ is shown in Figures 3 and 4. We can observe from Figure 3 that for a particular $q$, incorrect labeling decreases as $p$ becomes higher. We can also observe from Figure 4 that if $q = 0.5$ then the incorrect labeling is 50% because the complete information from the oracle is lost. For other values of $q$, we can see that the incorrect labeling decreases with increasing $d$.

We point out that 'same cluster' queries are not a good choice here, because of the symmetric nature of XOR due to which there is no gap between the expected numbers (contrary to the proof of Theorem 5) which we had exploited in the algorithm to a large extent.

Lastly, we show that Algorithm 1 can work without knowing the prior distribution and only with the knowledge of relative sizes of the clusters. The ground truth clusters can be adversarial as long as they maintain the relative sizes.

**Theorem 6.** *Suppose we have $n_i$, the number of elements with label $i$, $i = 0, 1, \ldots, k - 1$, as input instead of the priors. By taking a random permutation over the nodes while constructing the d-regular graph, Algorithm 1 will be $(1 - \delta)$-good approximation with $m = O(n \log \frac{k}{\delta})$ queries as $n \to \infty$ when we set $p_i = \frac{n_i}{n}$.*

The proof of this theorem is deferred to the appendix in the supplementary material.

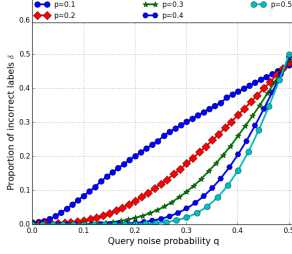
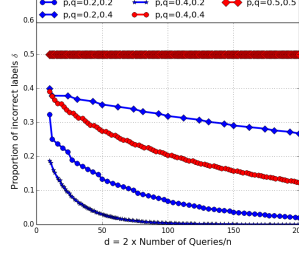
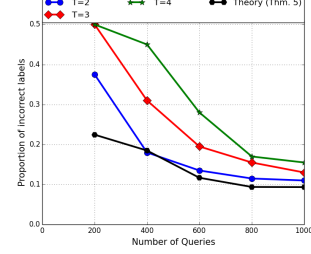

Figure 3: Recovery error for a fixed $p, d = 100$ and varying $q$

Figure 4: Recovery error for a fixed $p, q$ and varying $d$

Figure 5: Algorithm 1 on real crowdsourced dataset

# 4 Experiments

Though our main contribution is theoretical we have verified our work by using our algorithm on a real dataset created by local crowdsourcing. We first picked a list of 100 'action' movies and 100 'romantic' movies from IMDB (http://www.imdb.com/list/ls076503982/ and http://www.imdb.com/list/ls058479560/). We then created the queries as given in the querying scheme of Sec. 3.3 by creating a $d$-regular graph (where $d$ is even). To create the graph we put all the movies on a circle and took a random permutation on them in a circle. Then for each node we connected $\frac{d}{2}$ edges on either side to its closest neighbors in the permuted circular list. This random permutation will allow us to use the relative sizes of the clusters as priors as explained in Sec. 3.3. Using $d = 10$, we have $\frac{nd}{2} = 1000$ queries with each query being the following question: *Are both the movies 'action' movies?*. Now we divided these 1000 queries into 10 surveys (using SurveyMonkey platform) with each survey carrying 100 queries for the user to answer. We used 10 volunteers to fill up the survey. We instructed them not to check any resources and answer the questions spontaneously and also gave them a time limit of a maximum of 10 minutes. The average finish time of the surveys were 6 minutes. The answers represented the noisy query model since some of the answers were wrong. In total, we have found 105 erroneous answers in those 1000 queries. For each movie we evaluate the $d$ query answer it is part of, and use different thresholds $T$ for prediction. That is, if there are more than $T$ 'yes' answers among those $d$ answers we classified the movie as 'action' movie and a 'romantic' movie otherwise. The theoretical threshold for predicting an 'action' movie is $T = 2$ for oracle error probability $q = 0.105, p = 0.5$ and $d = 10$. But we compared other thresholds as well.

We now used Algorithm 1 to predict the true label vector from a subset of queries by taking $\tilde{d}$ edges for each node where $\tilde{d} < d$ and $\tilde{d}$ is even i.e $\tilde{d} \in \{2, 4, 6, 8, 10\}$. Obviously, for $\tilde{d} = 2$, the thresholds $T = 3, 4$ is meaningless as we always estimate the movie as 'romantic' and hence the distortion starts from $0.5$ in that case. We plotted the error for each case against the number of queries ($\frac{n\tilde{d}}{2}$) and also plotted the theoretical distortion obtained from our results for $k = 2$ labels and $p = 0.5, q = 0.105$. We compare these results along with the theoretical distortion that we should have for $q = 0.105$. All these results have been compiled in Figure 5 and we can observe that the distortion is decreasing with the number of queries and

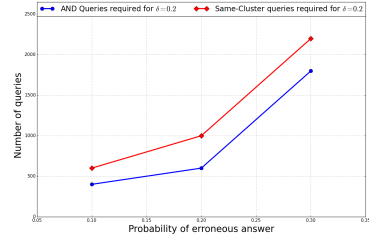

Figure 6: Comparison of 'same cluster' query with AND queries when both achieve 80% accuracy

the gap between the theoretical result and the experimental results is small for $T = 2$. These results validate our theoretical results and our algorithm to a large extent.

We have also asked 'same cluster' queries with the same set of 1000 pairs to the participants to find that the number of erroneous responses to be 234 (whereas with AND queries it was 105). This substantiates the claim that AND queries are easier to answer for workers. Since this number of errors is too high, we have compared the performance of 'same cluster' queries with AND queries and our algorithm in a synthetically generated dataset with two hundred elements (Figure 6). For recovery with 'same cluster' queries, we have used the popular spectral clustering algorithm with normalized cuts [24]. The detailed results obtained can be found in Figure 7 in the supplementary material.

**Acknowledgements:** This research is supported in parts by NSF Awards CCF-BSF 1618512, CCF 1642550 and an NSF CAREER Award CCF 1642658. The authors thank Barna Saha for many discussions on the topics of this paper. The authors also thank the volunteers who participated in the crowdsourcing experiments for this paper.

## Footnotes

[1]The difference between clustering and learning labels is that in the case of clustering it is not necessary to know the value of the label for a cluster. Therefore any unsupervised labeling algorithm will be a clustering algorithm, however the reverse is not true. In this paper we are mostly concerned about the labeling problem, hence our algorithms (upper bounds) are valid for clustering as well.

[2]Independent repetition of queries is also theoretically not interesting, as by repeating any query just $O(\log n)$ times one can reduce the probability of error to near zero.

[3]The compressed vector is not necessarily binary, nor it is necessarily smaller length.

[4]Here a typical set of labels is all such label vectors where the number of ones is between $np - n^{2/3}$ and $np + n^{2/3}$.

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
