[Supplementary Material]

# Semisupervised Clustering, AND-Queries and Locally Encodable Source Coding (Supplementary Material)

## Arya Mazumdar and Soumyabrata Pal

## A    Proof of Theorem 3

We will need the following definitions for this proof inspired by [6].

**Definition.** If the number of queries is $m$ and the number of input labels is $n$ then we define rate as the relative number of queries or $R = \frac{m}{n}$

**Definition.** The Rate-Distortion function $R(\delta)$ is the infimum of the feasible rates such that the scheme is $(1 - \delta)$-good.

**Definition.** The Distortion-Rate function $\delta(R)$ is the infimum of all $\delta$, for $(1 - \delta)$-good schemes, when the rate is $R$.

**Definition.** The set of reconstructed label vectors are called *codewords*. Since the rate is R, our problem is to define the querying scheme $Q : \{0, 1\}^n \to \{0, 1\}^{nR}$ and a recovery $\{0, 1\}^{nR} \to \{0, 1\}^n$. We have a bijective mapping from query answers $\mathbf{Y}$ to $\hat{\mathbf{X}}$. Hence the total number of possible codewords is $2^{nR}$.

*Proof of Theorem 3.* We are interested in finding a lower bound on the distortion that we will achieve if we use a rate $R(\tilde{\delta})$ for the model where $R(\tilde{\delta})$ is the minimum rate for distortion $\tilde{\delta}$ achieved optimally in the unconstrained case. Now suppose that the distortion achieved in the model at rate $R(\tilde{\delta})$ is $\delta = \tilde{\delta} + \epsilon$ and hence we want a lower bound on $\epsilon$ . Since our input labels are typical sequences and it is mapped to a unique codeword, the reconstructed sequence must be having a per symbol distortion of less than $\tilde{\delta} + \epsilon$ . We will be counting the number of label vector-codeword pairs $(S, T)$ where $S \in \{0, 1\}^n$ is a label vector and $T \in \{0, 1\}^n$ is the corresponding codeword for $S$. Let us allow a small extra distortion of $\gamma$. Now, we have 2 ways to count the number of possible pairs. Firstly, from the perspective of the codewords , the number of possible pairs will be $2^{nR}\text{Vol}(\tilde{\delta} + \epsilon + \gamma)$ where $\text{Vol}(\tilde{\delta} + \epsilon + \gamma)$ is the number of label sequences present in the ball of radius $n(\tilde{\delta} + \epsilon + \gamma)$ from a particular codeword. Since there might be repetitions hence we are overcounting and hence this value is definitely an upper bound on the number of pairs. Again we can try to see from the perspective of the label sequences. Now let us say that we have a label sequence $S$ and a corresponding compressed sequence $C$ and the codeword $T$ when no extra distortion is allowed. We will be trying to find out the number of other different codewords $S$ could have mapped to when this extra distortion is allowed. Let us take ball of $n\gamma$ around $S$ and take another label sequence in that ball and call it $\hat{S}$. Let the codeword it was initially mapped to be $\hat{T}$ Now,

$$|\hat{T} - S| \leq |\hat{T} - \hat{S}| + |\hat{S} - S| \leq n(\tilde{\delta} + \epsilon + \gamma)$$

Hence $\hat{T}$ is a possible candidate codeword for $S$ if we allow this extra bit of more distortion $\gamma$. Hence we want a lower bound on the number of possible different codewords that can be candidate codewords for $S$ when this extra distortion is allowed. Now we know that there is a bijective mapping from the compressed sequences to the codewords. Let $x$ denote the fraction of bits in $C$ that we can perturb. Then if, $nRx\Delta < n\gamma$ or $x < \frac{\gamma}{R\Delta}$ , then there exists label vectors mapped to those compressed sequences which will be within a ball of $n\gamma$ from $S$ and all the codewords corresponding to those perturbed compressed sequences must be different because of the bijective mapping. Since the total number of label sequences is $2^{nh(p)}$, the total number of pairs that can be calculated in such a way will be $2^{nh(p)+nRh(\frac{\gamma}{R\Delta})}$ which is a lower bound on the actual number of pairs. Since $\frac{1}{n}\log(Vol(\tilde{\delta} + \epsilon + \gamma)) = h(p) - R(\tilde{\delta} + \epsilon + \gamma)$ we have

$$R(\tilde{\delta}) - R(\tilde{\delta} + \epsilon + \gamma) \geqslant Rh(\frac{\gamma}{R\Delta})$$

Using the fact that $R(\delta) = h(p) - h(\delta)$, we have

$$h(\tilde{\delta} + \epsilon + \gamma) - h(\tilde{\delta}) \geq Rh(\frac{\gamma}{R\Delta}).$$

Now since entropy is a concave function of the distribution we must have $h(\tilde{\delta} + \epsilon + \gamma) \geq h(\tilde{\delta}) + (\epsilon + \gamma)h'(\tilde{\delta})$ where $h'(x) = \log \frac{(1-x)}{x}$ is the derivative of the binary entropy function. Plugging it into the formula, we have

$$\epsilon h'(\tilde{\delta}) \geq Rh(\frac{\gamma}{R\Delta}) - \gamma h'(\tilde{\delta})$$

Now we want the value of $\gamma$ in order to get the tightest lower bound. Hence, differentiating w.r.t $\gamma$ and setting it to 0 in order to maximize it, we will have $\gamma = \frac{R\Delta}{1+e^{\Delta h'(\tilde{\delta})}}$ Using this value of $\gamma$, we plug in the original equation and we get

$$\epsilon \geq -\frac{R\Delta}{1 + e^{\Delta h'(\tilde{\delta})}} + \frac{1}{h'(\tilde{\delta})} Rh(\frac{1}{1 + e^{\Delta h'(\tilde{\delta})}})$$

Expanding the entropy function we have

$$\epsilon \geq -\frac{R\Delta}{1 + e^{\Delta h'(\tilde{\delta})}} + \frac{R}{h'(\tilde{\delta})(1 + e^{\Delta h'(\tilde{\delta})})} \log(1 + e^{\Delta h'(\tilde{\delta})}) + \frac{Re^{\Delta h'(\tilde{\delta})}}{h'(\tilde{\delta})(1 + e^{\Delta h'(\tilde{\delta})})} \log(1 + \frac{1}{e^{\Delta h'(\tilde{\delta})}})$$

Now if use the facts that $\log(1 + e^{\Delta h'(\tilde{\delta})}) \geq \Delta h'(\tilde{\delta})$, $\log(1 + \frac{1}{e^{\Delta h'(\tilde{\delta})}}) \geq \frac{1}{e^{\Delta h'(\tilde{\delta})}}$ and $R(\tilde{\delta}) = h(p) - h(\tilde{\delta})$, we get that

$$\epsilon \geq \frac{h(p) - h(\tilde{\delta})}{h'(\tilde{\delta})(1 + e^{\Delta h'(\tilde{\delta})})}.$$

$\square$

## B  Exact Relation between $\delta, p, q, k$ and $d$ in Theorem 5

It is evident from the scheme that when the true label of $u$ is $i \neq 0$, $\mathbb{E}[N_{u,i}] = \frac{dq}{k-1} + dp_i\left(1 - \frac{qk}{k-1}\right)$ and when it is anything else, $\mathbb{E}[N_{u,i}] = \frac{dq}{k-1}$ and hence the threshold that we have described. Now let us calculate the probability of error under this scheme. It is evident that the error is symmetric $\forall i \neq 0$. Now there are two kinds of errors that are possible when the true label is $i$. The first type of error occurs when $N_{u,j} > C_j$ for some $j \neq i, 0$ and the second type of error occurs when $N_{u,i} < C_i$. Hence when the true label is $i \neq 0$, the first type of decoding error can be written as follows

$$\Pr(N_{u,k} > C_k | k \neq 0, k \neq i) = \sum_{j=\lceil C_k \rceil}^{d} \binom{d}{j}(\frac{q}{k-1})^j (1 - \frac{q}{k-1})^{d-j}$$

and hence taking a union bound over $k - 2$ labels, we have

$$Z_{i,1} \equiv \Pr(\cup_{k \neq i,0} N_{u,k} > C_k) \leq \sum_{k:k \neq i,0} \sum_{j=\lceil C_k \rceil}^{d} \binom{d}{j}(\frac{q}{k-1})^j (1 - \frac{q}{k-1})^{d-j}.$$

We define element $v$ to be a two-hop-neighbor of $u$ if there is at least one query which involved both the elements $u$ and $v$. For the second part of the error we have to condition on the number of two-hop neighbors of $u$ having label $i$ for $u$ when the true label is $i$. Let us denote the neighbor set of $u$ by $\text{Nbr}(u)$ and denote the number of nodes of label $i$ in its neighbor set as $\text{Nbr}_I(u)$. The probability of second type of error ($N_{ui} < \lceil C_i \rceil$) is

$$Z_{i,2} \equiv \sum_{\hat{k}} \Pr(\text{Nbr}_I(u) = \hat{k}) \Pr(N_{ui} < \lceil C_i \rceil \mid \text{Nbr}_i(u) = \hat{k})$$

$$= \sum_{\hat{k}} \binom{d}{\hat{k}} p_i^{\hat{k}} (1 - p_i)^{d-\hat{k}} \Pr(\hat{X}_u \neq X_u \mid \text{Nbr}_i(u) = \hat{k})$$

Now we take two cases for the value of $k$. When $\hat{k} \leq \lfloor C_i \rfloor$ then

$$\Pr(N_{u,i} < \lceil C_i \rceil \mid \text{Nbr}_i(u) = \hat{k}) = \sum_{i=0}^{\hat{k}} \binom{\hat{k}}{i}(\frac{q}{k-1})^i (1 - \frac{q}{k-1})^{\hat{k}-i}$$

Figure 7: Comparison of performance of 'same cluster' query with AND queries on a randomly generated dataset for varying probability of erroneous answers and varying number of queries. The AND querying methods performs well with higher probability of erroneous answers.

$$\sum_{j=0}^{\lfloor C_i \rfloor - \hat{k} + i} \binom{d - \hat{k}}{j} (\frac{q}{k-1})^j (1 - \frac{q}{k-1})^{d - \hat{k} - j}$$

and secondly when $\hat{k} \geq \lceil C_i \rceil$ then

$$\Pr(N_{u,i} < \lceil C_i \rceil \mid \mathrm{Nbr}_i(u) = \hat{k}) = \sum_{i=0}^{\lfloor C_i \rfloor} \binom{d - \hat{k}}{i} (\frac{q}{k-1})^i (1 - \frac{q}{k-1})^{d - \hat{k} + i}$$

$$\sum_{j=\hat{k} - \lfloor C_i \rfloor + i}^{\hat{k}} \binom{\hat{k}}{j} (\frac{q}{k-1})^j (1 - \frac{q}{k-1})^{\hat{k} - j}.$$

Now, if the true label of $u$ is 0

$$Z_0 = \cup_{k \neq 0} \Pr(N_{u,k} > C_k) = \sum_{k} \sum_{j=\lceil C_k \rceil}^{d} \binom{d}{j} (\frac{q}{k-1})^j (1 - \frac{q}{k-1})^{d - j}.$$

The total probability of error is going to be

$$\delta = p_0 Z_0 + \sum_{i \neq 0} p_i (Z_{i,1} + Z_{i,2}).$$

## C  Proof of Theorem 6

*Proof.* Let us consider $k = 2$. The proof can be extended for general $k$. We generate the $d$-regular graph on a $n$ nodes is the following way:

1. If $d = 2\ell$ is even, put all the vertices around a circle, and join each to its $\ell$ nearest neighbors on either side.

2. If $d = 2\ell + 1$ is odd, and $n$ is even, put the vertices on a circle, join each to its $\ell$ nearest neighbors on each side, and also to the vertex directly opposite.

Now suppose we are given $n_0$ and $n_1$. Let us consider all random permutations of these sets of points on a circle. Fixing a node $u$ of label 1(say), it becomes a random permutation on a line by making $u$ a reference point. The probability that there are exactly $k$ neighbors of $u$ having label 1 among the $d$ neighbors is

$$\binom{d}{k} \frac{\binom{n-d-1}{n_1-k-1}}{\binom{n-1}{n_1-1}} \approx \binom{d}{k} \frac{\Pi_{i=0}^{k-1}(n_1 - i)\Pi_{i=0}^{d-k-1}(n_0 - i)}{\Pi_{i=0}^{d-1}(n - i)} \leq \binom{d}{k} (\frac{n_1}{n})^k (\frac{n_0}{n - k})^{d-k}.$$

All the rest of the conditional probabilities used in the analysis in Appendix B stays the same. Now, $k$ is just a constant $\leq d$ and hence $n - k \approx n$ and hence asymptotically this distribution is equivalent to

the binomial distribution. Hence we can simply set $C = dq + \frac{dn_1}{2n}(1 - 2q)$ and then use the Algorithm 1. Our final probability of incorrect labeling is going to be $\delta = \frac{n_1}{n}P(X \neq \hat{X}|X = 1) + \frac{n_0}{n}P(X \neq \hat{X}|X = 0)$. Thus for large $n$ its behavior is exactly the same as with using priors. $\qquad\square$