[Reviews · NeurIPS 2017]

Reviewer 1



The authors study a clustering model where the learner is allowed to ask queries from an oracle and the goal is to find the target clustering. The queries can be same-cluster queries [or more generally, a fixed binary-valued function of the label of the input instances]. The authors study the query complexity of such models. The query-based clustering model is a well-motivated and relatively recent research area, with applications in record-deduplication, crowdsourcing, etc. The authors study the problem from the information theoretic perspective. In particular, they provide upper and lower bounds on the number of required queries to perfectly (or approximately) recover the target clustering. They also consider the case that the oracle is noisy. The information-theoretic tools that they use are interesting and new to ML community. Because no structural assumption is made about the problem (e.g., having a hypothesis class, etc.) the analysis is limited. In fact, the marginal distribution of the labels is the main parameter that the bounds are based on (aside from the noise level). Therefore, the studied setting is a bit far from the usual settings studied in learning theory. However, this is a first step, and can help in understanding the information-theoretic aspects of query-based clustering, especially the related lower-bounds.

Reviewer 2



This paper explores a clustering problem of which the goal is to reconstruct the true labels of elements from possibly noisy & non-adaptive query-answers, each associating only a few elements. It derives lower bounds on the number of queries required to reconstruct the ground-truth labels exactly or partially in a variety of scenarios. It also proposes a so-called "AND-queries" algorithm that applies to arbitrary k-cluster case, and evaluates the performance of the algorithm under a real data set. While the paper attempts to provide theoretical analyses for various scenarios as well as develops an algorithm together with a real-data simulation, it comes with limitations in presentation, clarity of the claimed results, and comparison to related works. Perhaps significant changes and possibly new developments are to be made for publication. See below for detailed comments. 1. (Repeating the same questions is not allowed): This constraint looks a bit restricted. Also it may degrade the performance compared to the no-restriction case, as allowing the same questions is one natural way to combat the noise effect. One proper way to justify the restricted model might be providing an extensive set of real-data simulations with a comparison to an algorithm tailored for the no-restriction case, and possibly showing a better performance of the proposed algorithm. If that is not the case, the no-restriction case is more plausible to investigate instead. The authors seem to concern about independence-vs-dependence issue for multiple answers w.r.t the same query, which might make analysis challenging. But this issue can be resolved focusing on a setting in which the same query is given to distinct annotators and therefore the answers from the annotators are assumed to be independent. 2. (Theorem 1): The noiseless setting with exact recovery was also studied in a recent work [Ahn-Lee-Suh] which also considers the XOR-operation ('same-cluster' queries) under a non-adaptive measurement setting. But the result in [Ahn-Lee-Suh] does not coincide with the one claimed in Theorem 1. For instance, when p=1/2 and \Delta is a constant, the query complexity in Theorem 1 reads m=n/log2, while the one in [Ahn-Lee-Suh] reads m=nlogn/\Delta. This needs to be clarified in details. Moreover, the proof of Theorem 1 is hard to follow. 3. (Theorem 3): The expression of the lower bound on \delta is difficult to interpret. In particular, it is difficult to infer the bound on m in terms of n and \delta, from which one can directly see the performance improvement due to the relaxed constraint, reflected in \delta. 4. (Theorem 4): The same comment given w.r.t. Theorem 3 applies here. In addition, how different is the upper bound from the lower bound in Theorem 3? Can be the gap unbounded? Again the proof is hard to follow. 5. (Figure 2): No comparison is made w.r.t. the same cluster query method – there is a comparison only to the same-cluster-query 'lower' bound. Wondering if there is a crossing point also when compared to the same-cluster-query 'method'. If that is the case, any intuition behind that? Actually the paper claims that the proposed approach outperforms the same-cluster query scheme, which cannot be supported from Figure 2. 6. (Algorithm 1): The proposed algorithm requires the prior knowledge on p or the relative sizes of clusters (in view of Theorem 6). How to obtain such information in practice? A proper justification might validate the practicality of the algorithm. Also the pseudo-code is hard to follow without relying on the detailed proof of Theorem 5, which is also difficult to grasp. 7. (Figure 3 & 4 & 5): No comparison is made w.r.t. the state of the arts such as the same-cluster-query method, which is crucial to support the claim made in several places: the AND-queries algorithm outperforms the same-cluster-query scheme. [Ahn-Lee-Suh] K. Ahn, K. Lee, and C. Suh, “Community recovery in hypergraphs,” Proceedings of Allerton Conference on Communication, Control, and Computing, 2016. ------------ (Additional comments) I have rebuttal. I found some comments properly addressed. For further clarification: 1. Still not clear as to whether repeating the same question is not helpful. Did the Nature paper show that it is not useful? Perhaps this can be clarified in a revision. 2. Re. the comparison to the same-query method: it would be clearer to include the detailed discussion as addressed in the response.

Reviewer 3



This is an interesting paper. I enjoyed reading it. In the paper, the authors consider the task of clustering data points via crowdsourcing wherein the problem is of recovering original labels of data points based on the answers to similar cluster queries. The authors propose bounds on the number of queries required to recover the labels considering noisy as well as noiseless answers for the queries. The main contribution of the paper is theoretical, along with some empirical simulations. The paper is well written, making it easy to understand the theoretical contributions for a general machine learning audience. Especially, the connection w.r.t. Information theory for deriving the bounds is explained nicely. I have the following concerns, in regards to improving the paper. (1) While the idea of AND queries is interesting theoretically, it seems to conflict with the original motivation for crowdsourcing. In the case of an AND query, effectively, the label of the cluster is also given as part of the answer rather than just answering if the data points of interest belong to same cluster or not. If this is the case indeed, as I interpret from the paper, why would one even need to ask those similarity queries ? Why not ask the label for a data point itself ? This is my primary confusion w.r.t. the contributions in this paper. (2) Is it possible to simplify the expressions for the bounds in Theorems 1,2,3,4, while keeping the essence. (3) In the experiments, it seems that only AND queries are considered. If so, the experimental study is in question unless the above point (1) is clarified. I am satisfied with the response of the authors for my questions above. So, voting for the acceptance of this paper.